# OpenReview forum: "Benchmarking Ethics in Text-to-Image Models: A Holistic Dataset and Evaluator for Fairness, Toxicity, and Privacy"
_ICLR.cc/2025/Conference — ICLR 2025 Conference Withdrawn Submission_

### Official Review · Reviewer_KB4v · 2024-11-02

**Soundness:** 2
**Presentation:** 2
**Contribution:** 2
**Rating:** 5
**Confidence:** 3

**Summary:**

This paper proposes a new benchmarking framework for Multi-modal LLMs (text-to-image) that utilizes a newly hand-crafted dataset and a modified and fine-tuned MLLM evaluator model. The authors start by providing a survey of existing works in this domain and point out their weak points, and propose a taxonomy of "problematic" outputs MLLMs could generate. The authors then collect/curate a new dataset of prompts that could potentially cause such outputs as well as corresponding images that can then be used to train an evaluator model. Using the dataset and evaluator model, the authors first show that the proposed evaluator shows more robust (stronger OOD) performance compared to existing models and then evaluate existing MLLMs on their safety, finding that many models can still be misguided easily.

**Strengths:**

- The topic is well-motivated and is timely where generated content is proving to become a real problem.
- Adding human annotations where necessary is a good touch. This gives more confidence to the actual human-alignment of the curated dataset.
- The cross-attention mechanism for joining modalities is a nice addition.

**Weaknesses:**

- More discussion of the actual 12 ethics categories in the main paper and not in the appendix would be nice. Even the discussion in Appendix C.1. reads like a longer version of the same content in the main paper. But I'm still left unclear on the exact definitions of the terms "fairness", "toxicity" and "privacy". Even if it was defined in previous literature, it would be nice to include.
- For a benchmark paper, the analysis section feels a bit short. For example, more insights into how exactly notable models behave for different ethics issue would be nice. Some of the discussion about ImageGuard could be replaced.
- The content organization can be improved. For example, the claim of KL being a better metric was not very well supported in the main paper, but I found a small section in Appendix F.1. that discusses this point.

**Questions:**

- Allowing mixed labels with human annotators is nice, but what happens if a minority annotator decides that an image is unsafe? Should the model still respect it? The mixed label cases on the right side of Fig. 2 makes me think so but I'm not sure.
- It's a little strange to place so much emphasis on the contrastive loss when in reality most of the objective (0.99 as reported) is actually autoregressive. The ablation test does indeed show that the addition of $\mathcal{L}_{con}$ improves the model performance, but what about the case where only contrastive loss is considered?
- Is the claim about variance true? Can the authors provide more examples of cases where "a model could have low variance but consistently underrepresent a particular group"?. More elaboration into this would be appreciated.
- Will ImageGuard predict the label using in natural language? How is it parsed? Can the prediction be a soft label (instead of just one category)?

---

### Official Review · Reviewer_XcAh · 2024-11-04

**Soundness:** 3
**Presentation:** 2
**Contribution:** 3
**Rating:** 6
**Confidence:** 3

**Summary:**

This paper proposes T2IEthics, a benchmark of ethical considerations in the context of text to image (T2I) models. In particular, the axes of evaluation include fairness, toxicity, and privacy, and there are 12 sub-categories in total. In addition, the paper proposes ImageGuard, a multimodal language model based evaluator for T2I models. Empirical evaluations on the cross-modularity attention (CMA) module in ImageGuard, and the benchmark experiments are presented.

**Strengths:**

The strength of the paper comes from the attempt to construct a benchmark for T2I models, which is interesting and timely. The proposed evaluator is MLLM-based, incorporating the proposed CMA module. The analysis of the evaluator itself is presented in detail, before presenting the benchmark results.

**Weaknesses:**

The paper can be further improved by considering possible re-organization of the material. The paper gets hard to parse from time to time. For example, the data collection pipeline is introduced in Section 3 and ImageGuard is introduced in Section 4. However, in the data collection pipeline (Section 3), there are multiple references to ImageGuard before it is even presented (later in Section 4). It would be beneficial to consider material reorganization to make the overall flow of the paper more natural and fluent.

Minor typo: line 200 opening quotation mark

**Questions:**

Detailed in the comment in "Weakness" section above.

---

### Official Review · Reviewer_EZhN · 2024-11-08

**Soundness:** 1
**Presentation:** 3
**Contribution:** 3
**Rating:** 3
**Confidence:** 4

**Summary:**

This paper makes three contributions towards safe and ethical image generation. First, this paper create and collate an ethical image and prompt benchmark assessing for a wide range of harms, including fairness, toxicity, and privacy. Second, this paper train a mllm using this dataset to classify potentially harmful images, and evaluate a range of popular image generation models. Third, this paper uses this benchmark to explore the effectiveness of two concept unlearning techniques at removing unethical and harmful concepts from T2I models.

**Strengths:**

-originality
This paper proposes two novel benchmarks that are more comprehensive of different harms categories than previous benchmarks. In addition, this paper creates a image moderation model that outperforms previous models.

-quality
This paper uses a a broader and more comprehensive set of harms categories than previous works. This paper also assesses a wide range of prominent models.

clarity
-Paper is generally well-written.

significance
Creating better benchmarks to assess harms, risks, and other ethical dimensions of T2I systems is deeply needed. This paper advances evaluations in this area, creating both a new dataset and a new assessment model.

**Weaknesses:**

This paper's primary flaw is the collection and human annotation of images.

1. Laion-5B contains CSAM and likely NCII, and should never be used.
2. Any research with humans requires an IRB. I cannot find mention of one in this paper.
3. What are harms to human annotators from viewing toxic content, how are those mitigated?
4. How were human annotators compensated?
5. What are limitations/biases of this approach?
6. Annotation schema is very unclear overall, how was each category defined, what instructions were presented to annotators?

The paper may be reworked to address these flaws, but I don't know if it is ready for publication with these issues.

**Questions:**

See weaknesses above.

**Details Of Ethics Concerns:**

No IRB, but used human annotators. Annotators were asked to annotate potentially toxic or hurtful images.

---

### Official Review · Reviewer_TrLT · 2024-11-08

**Soundness:** 2
**Presentation:** 1
**Contribution:** 3
**Rating:** 3
**Confidence:** 3

**Summary:**

This work proposes a framework for evaluating text-to-image (T2I) models with respect to properties related to fairness, toxicity, and privacy. The major components of the study are (1) a taxonomy of specific categories of concepts that their framework aims to identify, (2) the curation of a set of prompts that can be used for adversarial testing with respect to the defined categories, (3) the creation of “ImageGuard”, a vision-language model that predicts the presence of the categories of interest in image-text pairs, and (4) empirical study of existing T2I models under the proposed framework. In their empirical study, they find that ImageGuard performs well compared to comparator models and demonstrate the capability of the framework to compare performance of T2I under the dimensions identified.

**Strengths:**

* The problem that this work addresses is important. There is currently an unmet need for frameworks for evaluating various forms of bias and harm in T2I models.
* This work makes concrete progress on addressing the stated aims through the creation of datasets for benchmarking and the release of ImageGuard for scalable evaluation without the use of human annotators. The datasets and benchmarks released are varied in scope and each may be independently of interest for reuse in future work.
* The scope of the empirical evaluation is broad in that they compare ImageGuard to ten other models and they evaluate twelve T2I models under the proposed framework, demonstrating the use of the approach is indeed scalable.

**Weaknesses:**

* The work generally over-claims its contributions, which has the effect of weakening the paper as a whole. Specifically, the paper refers to the proposed framework and approach as “comprehensive” in its coverage of all relevant concepts related to ethical aspects of T2I models. However, the scope of the issues that the proposed approach covers is still relatively narrow and not comprehensive, even if it improves upon the coverage of comparable prior works. For example, the space of relevant fairness concerns is much vaster than simply measurement of the diversity of generated outputs with respect to a small number of race, gender, and age groups, and requires some grounding in the sociotechnical contexts in which T2I systems are intended to be used. This work shows no awareness of these broader fairness-related issues. Similarly, with respect to privacy, while they cover generation of identifiable documents, public figures, and intellectual property violation, they do not cover other well-established relevant notions of privacy, including differential privacy. The proposed categories of “toxicity” (i.e., illegal activities, humiliation, distributing, hate, sexual, and violence) are reasonable at face value, even if “toxicity” is not the right nomenclature for this upper-level category, but it is again disingenuous to claim that these categories are comprehensive, or even novel, as several related works have proposed more detailed and specific taxonomies of harms to use to define criteria for identifying harms of generative models (e.g., Shelby 2023, Weidinger 2023). Furthermore, it is inappropriate to claim that privacy, fairness, and toxicity are comprehensive in their coverage of ethical issues.
  * A specific example of some of this overclaiming occurs in lines 125-130, “Despite these contributions […]”.
* The approach to reasoning about fairness is not appropriately motivated or described. In section 3.3., they state that “traditional metrics like accuracy or variance are insufficient to capture true fairness”, with the implication that the proposed normalized KL divergence is sufficient to measure “true fairness”. However, the work does not clearly state what they are trying to measure nor motivate why this is the right notion of fairness to measure. I believe the intent is to measure whether the generated images correspond to a uniform distribution over the groups defined for a particular sensitive attribute (race, gender, age), but this is not stated. To be clear, I do believe that the normalized KL divergence with respect to a uniform distribution is a reasonable way to measure that. As mentioned above, there are a number of different properties relevant to reasoning about fairness (see e.g., Barocas 2023), and the proposed approach is only one of many. It is inappropriate to claim that this approach corresponds to “true fairness”.
* On several occasions, the work inappropriately describes the artifacts generated as a part of this work as “ethical”. For example, they refer to datasets generated as “ethical data” (line 190), the prompts collected as “ethical prompts” (section 3.2.1), and the benchmark as an “ethical benchmark” (line 210). It would be more accurate to say that these artifacts enable identification of particular forms of violation of ethical principles, but they are not themselves “ethical”.
* The approach to defining categories for fairness assessment is potentially ethically problematic. There are a few dimensions to this issue. First, the taxonomy of categories for each attribute is limited to a small number of categories that are not comprehensive. For instance, the work enforces a gender binary and defines an arbitrary set of racial categories (Caucasian, African, Indian, Asian and Latino). To construct the datasets, human raters then annotate for these categories based on their perception of the image, and these labels are then encoded in the training of the ImageGuard model. The final result is a model that classifiers gender and race from images. For gender, this is problematic due to erasure and misclassification of non-binary and transgender individuals (Scheuerman 2019). With respect to race, while the incompleteness of the particular taxonomy presented is perhaps unavoidable, it is potentially problematic that the end result is a classification system based on arbitrary categories that cannot be faithfully inferred from images (Hanna 2020).

References
1. Shelby, Renee, et al. "Sociotechnical harms of algorithmic systems: Scoping a taxonomy for harm reduction." Proceedings of the 2023 AAAI/ACM Conference on AI, Ethics, and Society. 2023.
2. Weidinger, Laura, et al. "Sociotechnical safety evaluation of generative ai systems." arXiv preprint arXiv:2310.11986 (2023).
3. Barocas, Solon, Moritz Hardt, and Arvind Narayanan. Fairness and machine learning: Limitations and opportunities. MIT press, 2023.
4. Scheuerman, Morgan Klaus, Jacob M. Paul, and Jed R. Brubaker. "How computers see gender: An evaluation of gender classification in commercial facial analysis services." Proceedings of the ACM on Human-Computer Interaction 3.CSCW (2019): 1-33.
5. Hanna, Alex, et al. "Towards a critical race methodology in algorithmic fairness." Proceedings of the 2020 conference on fairness, accountability, and transparency. 2020.

**Questions:**

* A few minor points to clarify in the text:
  * Line 247: more detail is needed to define the “traditional safety rate”
  * A “perceive sampler” is not defined or cited (Figure 3 caption, section 4.3)
  * In section 5.2., the framing around fairness is flipped. The text implies that more fairness is a negative thing and that less fairness is a positive thing. Consider framing around more or less “fairness violation”.
  * It is unnecessary and inaccurate to say that the F1 score is a comprehensive measure of the performance of a classifier (line 367).

* Why does the correlation analysis of ImageGuard with human annotators (Table 5) not include all of the models listed in Table 3? Why is CLIP-L the right comparator?
* In the description of SMID (line 361-363), what does the “moral value” refer to?
* The paper would be improved if some elaboration on how the categories of the toxicity taxonomy were defined.

**Details Of Ethics Concerns:**

[Copied from weaknesses section]

The approach to defining categories for fairness assessment is potentially ethically problematic. There are a few dimensions to this issue. First, the taxonomy of categories for each attribute is limited to a small number of categories that are not comprehensive. For instance, the work enforces a gender binary and defines an arbitrary set of racial categories (Caucasian, African, Indian, Asian and Latino). To construct the datasets, human raters then annotate for these categories based on their perception of the image, and these labels are then encoded in the training of the ImageGuard model. The final result is a model that classifiers gender and race from images. For gender, this is problematic due to erasure and misclassification of non-binary and transgender individuals (Scheuerman 2019). With respect to race, while the incompleteness of the particular taxonomy presented is perhaps unavoidable, it is potentially problematic that the end result is a classification system based on arbitrary categories that cannot be faithfully inferred from images (Hanna 2020).

---

### Official Review · Reviewer_DGBi · 2024-11-09

**Soundness:** 2
**Presentation:** 2
**Contribution:** 2
**Rating:** 5
**Confidence:** 5

**Summary:**

The authors have essentially packaged 3 ethics related modules in this paper specifically covering text-to-image generative models.
a) A multi-level ethical evaluation framework/ benchmark termed "T2IEthics"
b) An image content moderator using MultimodalLLMs called "ImageGuard"
c) An ethical-evaluation leaderboard that covers 12 near-SoTA text-to-image diffusion models

**Strengths:**

This paper's biggest strength lies in the sheer volume of effort invested. The authors have introduced a substantial ethics taxonomy, used Multimodal-LLMs to create an "ImageGuard" and empirically generated a 12 model scorecard that covers pretty much all the SoTA models available.

**Weaknesses:**

My main concern about the paper is that the authors have focused on comprehensiveness and volume of effort rather than contributing a truly novel idea or angle of scholarship that can help us implement better guardrails around T2I models. This is especially relevant now given that the very institution of an academic paper is being meaningfully challenged and disrupted by SoTA LLM-based frameworks that can output both code and prose while also running empirical experiments on the human's behalf.

I'd like the authors to specifically consider this scenario:

A software engineer is extremely adroit at prompting LLMs. They have now been entrusted by their manager to deploy a 'safe & ethical' T2I app by just shopping around for open-source models and adding an AI-ethics boilerplate layer on top of them. This includes fabricating a typology of shortcomings, a probe-dataset, an appropriate metric to measure the shortcomings and a fancy scoring-dashboard to capture the scores. Assume they have access to a fair amount of free credits to use any of the commercial frameworks such as GPT-x, Claude-y or Perplexity-z and are expected to produce a report justifying their choice. How does their report meaningfully look worse compared to your scholarly efforts?
When I tried prompting one of these commercial frameworks with the following prompt: "Give me a comprehensive hierarchical ethics framework covering all the possible shortcomings and checks that will be used to audit text-to-image models so that they are real-world deployable", I received a far more comprehensive landscape of tests than provided in Table 9.
Given that we now live in the post-GPT-3 era, it serves to examine as researchers as to what facets of the scholarship we are producing is genuinely not GPT-fiable?

Other specific critiques:

A: Grammar and tonality:

I'd implore the authors to refrain from critiques that are hand-wavy. For example, in the abstract section, you state : "Existing safety benchmarks are limited in scope, lacking comprehensive coverage of critical ethical aspects such as detailed categories of toxicity, privacy, and fairness, and often rely on inadequate evaluation techniques".

Specific changes:

Line 51: Please fix the grammar: "The rapid rise of text-to-image (T2I) models have been used to generate high-quality, realistic images from text descriptions from various domains and art styles."

Line 108: Please fix the capitalization and grammar:
"For evaluation, W(w)e introduce image ethical evaluator that greatly surpass(es) current
prevailing method, CLIP, enhancing the accuracy and reliability of ethical evaluations."

Line 1107: "Distance metric is symmetric, so it treats differences from both distributions
equally". Please rephrase this sentence. '... treats differences from both distributions' is a non-rigorous and non-scholarly claim.

Line 489: "The higher Cohen’s kappa scores indicate that ImageGuardaligns" (add space)

B: Please add a few lines that motivated the formulation of Table-9 categories.

C: Appendix-D: I'd remove this section as the material presented in available in most undergraduate level information theory textbooks.

**Questions:**

Deontological v/s Consequentialist ethics:
The authors have chosen to focus on deontological  "hot-button axes" of ethical shortcomings such as those pertaining to fairness/ toxicity/ privacy in Table 9 and not considered one of the most worrisome downstream repercussions of T2I models: Fake misinformation AI slop.

Question:
Can the authors label whether these prompts ought to pass the ethics tests for T2I generation?

" A realistic photograph of a cute baby alligator licking on a strawberry ice-cream cone"
"A realistic image of moon aligned right next to Saturn above the Eiffel tower"
"A hyper-realistic flying saucer floating above the Pyramid of Giza"
" A realistic photograph of an African inventor kid repairing a partially constructed submarine"

Context: The images produced by these prompts are flooding social media platforms tricking non-tech-savvy users to think these are real or worse coaxing them in to scam payments where they think they are helping this genius innovator kid fulfill his dreams of building a submarine.
In lieu of developing prescriptive top-down  ethical lookup-tables (as in Table 9), I'd implore the authors to survey the landscape of harms being already caused and reported and develop the ethical-axes bottom-up.

Question-2: Would this prompt be considered ethical?
 " A realistic photo capturing a visibly non-binary hispanic person modeling with green jacket and red leather boots".
Context: This image risks being used by an ad agency to print ads without needing to actually hire a human from the minoritized population that they are purportedly centering and representing in the ad.

---

### Note · Authors · 2024-11-15

I have read and agree with the venue's withdrawal policy on behalf of myself and my co-authors.